# APPLIED TIME-SERIES TRANSFER LEARNING

**Nikolay Laptev, Jiafan Yu & Ram Rajagopal**
Department of Electrical Engineering
Stanford University
Stanford, CA, USA
{`nlaptev,jfy,ramr`}@stanford.edu

## ABSTRACT

Reliable and accurate time-series modeling is critical in many fields including energy, finance, and manufacturing. Many time-series tasks, however, suffer from a limited amount of clean training data resulting in poor forecasting, classification or clustering performance. Recently, convolutional neural networks (CNNs) have shown outstanding image classification performance even on tasks with small-scale training sets. The performance can be attributed to transfer learning through ability of CNNs to learn rich mid-level image representations. For time-series, however, no prior work exists on general transfer learning. In this short paper, motivated by recent success of transfer learning in image-related tasks, we are the first to show that using an LSTM auto-encoder with attention trained on a large-scale timeseries dataset with pre-processing we can effectively transfer time-series features across diverse domains.

Accurate time-series modeling is critical for load forecasting, financial market analysis, anomaly detection, optimal resource allocation, budget planning, and other related tasks. While time-series modeling has been investigated for a long time, the problem is still challenging, especially in applications with limited or noisy history (e.g., holidays, sporting events) where practitioners are forced to use adhoc machine learning approaches achieving poor performance (Wu & Olson, 2015).

Transfer learning (Pan & Yang, 2010) can address this problem. In transfer learning, we first train a base network on a base dataset and task, and then we repurpose the learned features, or transfer them, to a second target network to be trained on a target dataset and task. Recent findings of Bengio (2012) show preliminary results of successfully using transfer learning on images. Motivated by these and many other (Yosinski et al., 2014; Huang et al., 2013; Karpathy et al., 2014; Oquab et al., 2014b) results we investigate if transfer learning also applies to time series.

Transfer learning involves the concepts of a task and of a domain. A domain $D$ consists of a marginal probability distribution $P(X)$ over the feature space $\mathcal{X} = \{x_1, ..., x_n\}$. Thus, given a domain $\mathcal{D} = \{\mathcal{X}, P(\mathcal{X})\}$, a task $T$ is composed of a label space $\mathcal{Y}$ and a conditional probability distribution $P(Y|X)$ that is usually learned from training examples consisting of pairs $x_i \in \mathcal{X}$ and $y_i \in \mathcal{Y}$. Given a source domain $\mathcal{D}_S$ and a source task $\mathcal{T}_S$ as well as a target domain $\mathcal{D}_T$ and a target task $\mathcal{T}_T$, transfer learning aims to learn the target conditional probability distribution $P(Y_T|X_T)$ in $\mathcal{D}_T$ from the information learned from $\mathcal{D}_S$ and $\mathcal{T}_T$. In this workshop paper we apply transfer learning to a time-series domain and apply it to cases where $\mathcal{X}_S \neq \mathcal{X}_T$ and $P(Y_S|X_S) \neq P(Y_T|X_T)$ (e.g., target domains with limited training data, different tasks and different time-series classes).

We propose to learn the nonlinear mapping in time series forecasting with the help of attention-based LSTM auto-encoder. We first pre-process the training data by detrending, deseasoning and window-normalizing. Using an attention mechanism aims to identify the part of the time-series that the model should focus on, therefore allowing the model to switch its focus based on the current input and what the model produced so far. Using time-series pre-processing and LSTM autoencoder with attention allow us to successfully learn and transfer time-series features to a wide verity of tasks.

In this workshop paper, we show that using a model based on a long short term memory (LSTM) (Hochreiter & Schmidhuber, 1997) auto-encoder with attention (Vaswani et al., 2017) trained on a large-scale time-series dataset with pre-processing we can effectively transfer features across diverse domains. The use of an LSTM-based model is motivated by its continued success in modeling

sequences[1]. The use of the attention mechanism is used to improve embedding efficiency through attending to specific parts of the input. During training, we found that pre-processing the time-series by detrending, deseasoning and normalizing is critical to deal with diverse target domains. We systematically demonstrate that the LSTM with auto-encoder with attention has potential in feature extraction, classification, disaggregation, and forecasting, by testing on multiple cross-domain datasets.

**Abstract Feature Extraction from LSTM for Time Series Classification** We first show that the proposed attention-based auto-encoder on LSTM can successfully extract generalized features from time series. Such features are used for unlabelled time series classification and we show substantial improvements on classification accuracy over only using traditional features/statistics of time series. We use the standard UCR time-series classification dataset (Chen et al., 2015). The accuracy of the classifier is shown in Figure 1a. The $y$-axis is the accuracy of the classifier. Each bar shows the accuracy for a certain category. The classifier is trained using either only standard time-series features (e.g., variance, seasonality, trend), or the abstract features extracted by the proposed attention-based auto-encoder. Besides the difference of features, all other settings (machine learning model, training/testing size, etc.) are identical for two classifiers. The top of the blue bar is the accuracy for classifier with standard features, and the top of the red bar is the accuracy for classifier with auto-encoder extracted features. Hence, the length of the red bar demonstrates a great boost in performance relative to the baseline method of using standard time-series features.

**Transfer Learning for Time Series Disaggregation** We show transferability of the learned time-series features for disaggregation. In particular, we aim to estimate individual appliance usage given an aggregate power consumption. We compare the disaggregation performance with and without attention-based auto-encoder framework, by using Pecan Street dataset (Pecan Street Inc., 2017), which contains hourly measurement of power consumption of homes and individual appliances from 345 homes with each having complete record for at least one appliance, mainly located in Austin, Texas, in 2016. We use the pre-trained model on a large, generalized dataset, then fine-tune the last prediction layer for each individual time series on the disaggregation task. The individual time series is different from any time series in the generalized dataset. The disaggregation performance is shown in Figure 1b. The layout of Figure 1b is similar with the layout of Figure 1a. It is shown that using transfer learning, the accuracy for disaggregation of different appliance types is substantially increased.

**Transfer Learning for Time Series Forecasting** We also demonstrate the transferability of time series forecasting models. To train the forecasting model with transfer learning, we also first use the pre-trained LSTM based forecasting model on a large dataset. Then, for any individual time series (not from the large dataset), we fine tune the fully connected layers of the LSTM model based on the individual data. We use a large-scale electricity consumption dataset containing 116,000 anonymous residential electricity loads at hourly granularity from Pacific Gas and Electric Company (PG&E dataset). The 116,000 time series are from 13 different climate zones and are with large diversity (Kwac et al., 2014). We also test the transferability of time-series forecasting on standard M3 dataset. The result is shown in Figure 1c and Figure 1d. The $y$-axis is the symmetric mean absolute percentage error(SMAPE). We compare the performances for both in-sample training data, and out-of-sample test data. The constant improvements of using transfer learning (red bars) over not using transfer learning (blue bars) show competitive performance even on short time-series such as those in the M3 dataset.

In summary, we present an attention-based auto-encoder architecture for time-series feature extraction, trained on a large scale pre-processed time-series dataset with plans to be made publically available as a pre-trained model for time-series applications similar to the image equivalent that exists today (Oquab et al., 2014a). We also show the transferability of the learned time-series features to classification, disaggregation, and forecasting tasks. We are currently focusing on providing this work as a service for practitioners to use as an online tool for time-series feature generation or as an offline pre-trained model to be used as a prior for time-series machine learning tasks.

---

[1]Laptev et al. (2017) show that an LSTM forecasting model is able to outperform classical time series methods in cases with long, interdependent time series.

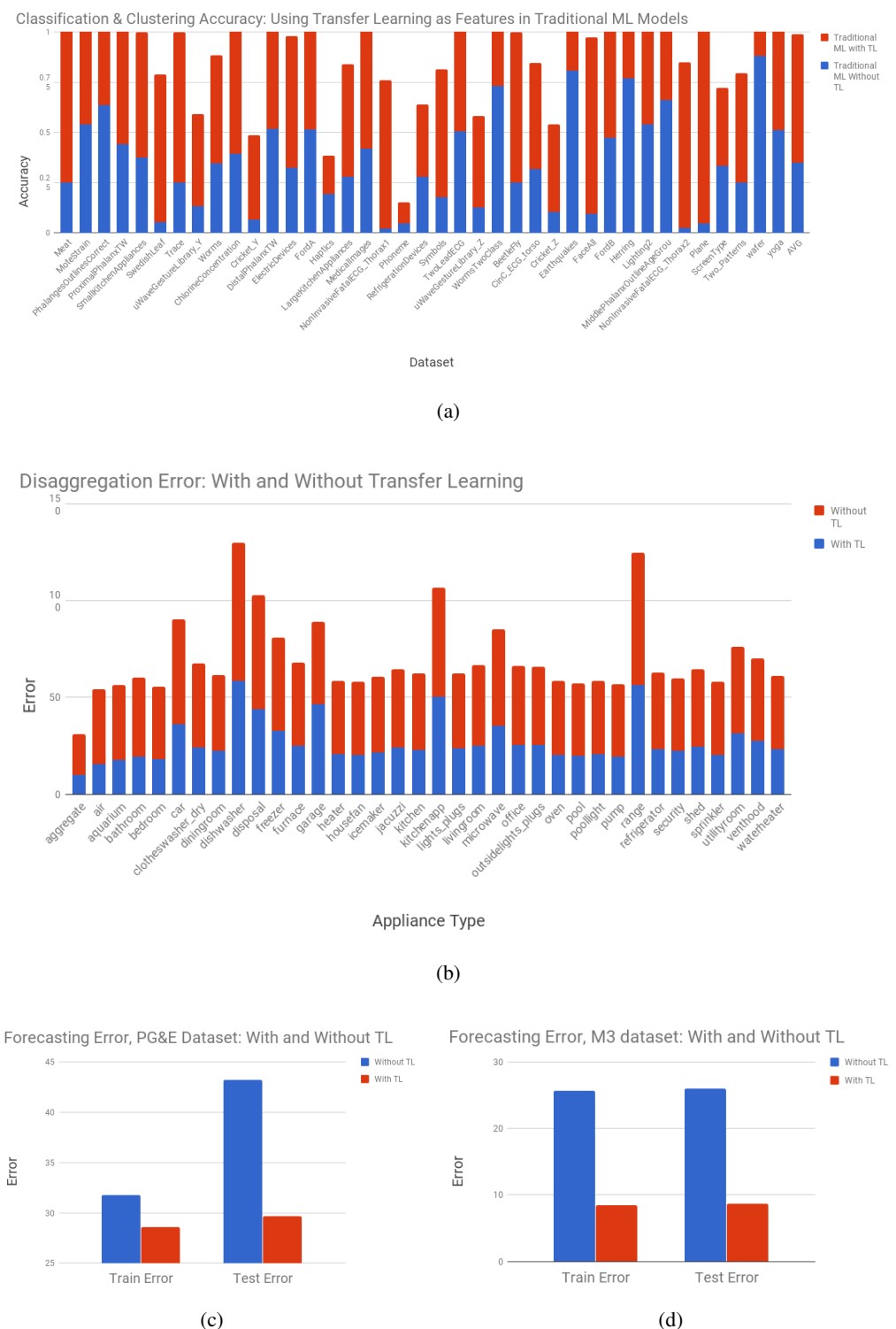

(a)

(b)

(c)

(d)

Figure 1: **(a)** Performance comparison of classification/clustering task on the UCR dataset by directly using the learned features in traditional machine learning models. **(b)** Transfer learning applied to the disaggregation tasks. **(c),(d)** Forecasting task evaluation by leveraging transfer learning on new datasets.

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
