# OpenReview forum: "Applied timeseries Transfer learning"
_ICLR.cc/2018/Workshop — Reject_

### Official Review · AnonReviewer2 · 2018-03-04
**Solid - nothing too surprising**

**Rating:** 6
**Confidence:** 3

**Review:**

This is a solid workshop paper. It is well-written; motivation, approach and experimental evaluation are mostly clear. The description of the attention mechanism is a bit vague ("identify the part of the time-series that the model should focus on"); I think that paragraph could be improved. Moreover, what is the unit of errors in Figure 1 b)?

In terms of originality or significance, this seems to be a solid proof of concept. It's good to see that transfer learning works for time series using LSTM models. On the other hand, this is not too surprising, either; overall the approach seems straight forward.

In summary
+ well-written
+ solid experiments demonstrating the approach on different datasets
- somewhat limited in terms of originality

---

### Official Review · AnonReviewer3 · 2018-03-09
**Interesting paper but in my view not good fit for workshop track**

**Rating:** 5
**Confidence:** 5

**Review:**

This paper proposes a method to transfer features across diverse domains. In order to do that, they trained an LSTM auto-encoder with attention on large-scale time series. Then learned time-series features are used for transfer learning on different tasks such as classification, disaggregation, and forecasting tasks. As mentioned in the paper, preprocessing such as detrending, deseasoning, and normalizing of time-series data play an important role to deal with diverse target domains.

Even though this paper shows interesting results, I am not sure if there is enough contribution either on the model side or even on the task side. In my view, this paper belongs to application paper category (and it is a solid application paper) rather than late-breaking development, very novel ideas, or position paper which are the main focuses of the workshop.

As a result, I am not convinced this paper is a good fit for the workshop track.

---

### Official Review · AnonReviewer1 · 2018-03-10

**Rating:** 3
**Confidence:** 4

**Review:**

This paper proposes an LSTM based model for transfer learning in time series data. It is not entirely clear to me what the experiment setting is. For example, in the time series classification with UCR datasets, what is the source and target domains for these plots? What method is "traditional ML without TL"? Why is this an autoencoder instead of just an encoder model? The writing needs to be improved so that people can understand exactly what the model is and what the experiment settings are.

---

### Public Comment · (anonymous) · 2019-03-06
**Question about classification results**

Hi,

In your classification results the transfer learning model used extracted features from a deep learning model. But non-transfer model just use traditional ML method with hand-crafted features. Have you compared the transfer learning model with LSTMs trained directly on the target data?

Is it possible that the transfer learning method has better performance just because it utilizes representation learning while the non-transferred one does not?

---

### Decision · Program_Chairs · 2018-03-20
**ICLR 2018 Workshop Acceptance Decision**

**Decision:**

Reject

**Comment:**

Based on the reviews, this paper has not been accepted for presentation at the ICLR workshop. However, the conversation and updates can continue to appear here on OpenReview.